# Vtc4 Promotes the Entry of Phagophores into Vacuoles in the *Saccharomyces cerevisiae* Snf7 Mutant Cell

**DOI:** 10.3390/jof9101003

**Published:** 2023-10-11

**Authors:** Xiaofan Chen, Yongheng Liang

**Affiliations:** Key Laboratory of Agricultural Environmental Microbiology of Ministry of Agriculture, College of Life Sciences, Nanjing Agricultural University, Nanjing 210095, China; 10319117@njau.edu.cn

**Keywords:** autophagy, ESCRT, phagophore, Snf7, vacuole, Vps21, Vtc4, VTC complex

## Abstract

Endocytosis and autophagy are the main pathways to deliver cargoes in vesicles and autophagosomes, respectively, to vacuoles/lysosomes in eukaryotes. Multiple positive regulators but few negative ones are reported to regulate the entry of vesicles and autophagosomes into vacuoles/lysosomes. In yeast, the Rab5 GTPase Vps21 and the ESCRT (endosomal sorting complex required for transport) are positive regulators in endocytosis and autophagy. During autophagy, Vps21 regulates the ESCRT to phagophores (unclosed autophagosomes) to close them. Phagophores accumulate on vacuolar membranes in both *vps21*∆ and ESCRT mutant cells under a short duration of nitrogen starvation. The vacuolar transport chaperon (VTC) complex proteins are recently found to be negative regulators in endocytosis and autophagy. Phagophores in *vps21*∆ cells are promoted to enter vacuoles when the VTC complex proteins are absent. Phagophores are easily observed inside vacuoles when any of these VTC complex proteins (Vtc1, 2, 4, 5) are removed. However, it is unknown whether the removal of VTC complex proteins will also promote the entry of phagophores into vacuoles in ESCRT mutant cells under the same conditions. Snf7 is a core subunit of ESCRT subcomplex III (ESCRT-III), and phagophores accumulate in *snf7*∆ cells under a short duration of nitrogen starvation. We used green fluorescence protein (GFP) labeled Atg8 to display phagophores and FM4-64-stained or Vph1-GFP-labeled membrane structures to show vacuoles, then examined fluorescence localization and GFP-Atg8 degradation in *snf7*∆ and *snf7*∆*vtc4*∆ cells. Results showed that Vtc4 depletion promoted the entry of phagophores in *snf7*∆ cells into vacuoles as it did for *vps21*∆ cells, although the promotion level was more obvious in *vps21*∆ cells. This observation indicates that the VTC complex proteins may have a widespread role in negatively regulating cargos to enter vacuoles in yeast.

## 1. Introduction

The yeast vacuole is a hub for degradation and signal transduction of many pathways [1]. Endocytosis and macroautophagy are typical pathways to deliver cargos to vacuoles in yeast and to lysosomes in mammalian cells [2]. Endocytosis is a process to actively internalize molecules and surface proteins via an endocytic vesicle, which may end up in lysosomes/vacuoles [3]. Macroautophagy is a process to engulf cargoes in double-membrane phagophores (precursors of autophagosomes or unclosed autophagosomes), which are sealed to form autophagosomes and to fuse with lysosomes/vacuoles to degrade the enclosed materials for recycling purposes [4]. In contrast to macroautophagy, eukaryotic cells directly degrade a wide range of autophagic cargoes through microautophagy, which has emerging mechanisms and functions and has drawn much attention to the autophagy field [5,6]. Both endocytosis and macroautophagy (hereafter referred to as autophagy) are associated with physiopathology and cell survival under different stress conditions. Both vesicles from endocytosis and autophagosomes from autophagy can fuse with lysosomes/vacuoles. These fusion steps are positively regulated by the fusion machinery, including but not limited to Rab7/Ypt7, homotypic fusion and vacuole protein sorting (HOPS) tethering complexes and soluble N-ethylmaleimide sensitive fusion protein (NSF) attachment receptors (SNAREs) [7,8,9]. The absence of any of these regulators inhibits the fusion process.

The yeast endosomal Rab5 GTPase Vps21 [10] and the endosomal sorting complex required for transport (ESCRT), which is composed of conserved multiple protein complexes, ESCRT-0, ESCRT-I, ESCRT-II, ESCRT-III, and a number of accessory proteins (e.g., the Vps4 ATPase) [11], play roles before the fusion step with vacuoles in both endocytosis and autophagy pathways [9,12]. The absence of Vps21 and ESCRT subunits results in phagophore accumulation under a short duration of autophagy induction [12,13,14]. However, these phagophores entered vacuoles under a prolonged duration of nitrogen starvation [15]. Previously, only closed autophagosomes were supposed to enter vacuoles/lysosomes [16]. However, recently, it became clear that phagophores can also fuse with lysosomes or vacuoles in mutant yeast and mammalian cells, although the molecular machinery is unknown [4,15,17]. Failure of autophagosome biogenesis and phagophore closure results in an interrupted autophagy process, which compromises homeostasis and leads to various diseases, including metabolic disorders, neurodegeneration and cancer [18]. The fusion of phagophores with lysosomes/vacuoles partially restores cargo degradation and recycling, maintaining autophagy to a certain degree to resist stresses [4,15]. In response to nitrogen starvation in the presence of a poor carbon source, diploid cells of *S. cerevisiae* undergo meiosis and package the haploid nuclei produced in meiosis into a tetrad with four spores [19]. The *vps21*∆/*vps21*∆ diploid cells are with phagophore accumulation of autophagy defect during a short duration but with near-complete autophagy and sporulation ability during a long duration of nitrogen starvation, which are different from the loss of sporulation ability in autophagy completely impaired mutant (*atg1*∆/*atg1*∆ and *pep4*∆/*pep4*∆) diploid cells, providing solid evidence for the physiological roles of phagophore–vacuole fusion [15].

The vacuolar transport chaperon (VTC) complex is a novel family of yeast chaperons (including Vtc1-5) mainly involved in the synthesis and transfer of polyP to vacuole, the distribution of V-ATPase and other membrane proteins, microautophagy, and membrane trafficking [20,21,22,23]. VTC complex proteins are negative regulators for the entry of misfolded glycosylphosphatidylinositol (GPI)-anchored proteins into vacuoles with microautophagy [24]. We recently found that in *vps21*∆ cells, the depletion of most Vtc proteins (four of five) promoted the entry of phagophores into vacuoles, while the depletion of Ypt7 and Vam3 in *vps21*∆ cells inhibited this process, by determining the changes in GFP-Atg8 vacuolar localization and degradation [15]. However, it is unknown whether VTC complex proteins have a general negative regulatory role in promoting proteins, complexes, vesicles, or bigger membrane structures into vacuoles. We reported that the absence of Vps21 and ESCRT subunits in yeast cells resulted in similar phagophore accumulation during autophagy [12]. We were curious to know whether the depletion of VTC complex proteins also promoted the entry of phagophores in ESCRT mutant cells into vacuoles. Snf7 is a core subunit of ESCRT-III, and phagophores accumulate in *snf7*∆ cells under a short duration of nitrogen starvation [12]. We removed a representative VTC complex protein Vtc4 from *snf7*∆ cells as from *vps21*∆ cells and similarly examined phenotypes [15]. We examined the vacuolar localization and degradation of phagophores in *snf7*∆ and *snf7*∆*vtc4*∆ cells, in which phagophores were shown with green fluorescence protein (GFP) labeled Atg8, and vacuoles were stained with lipophilic dye FM4-64 or labeled with Vph1-GFP. We used *vps21*∆ and *vps21*∆*vtc4*∆ cells as controls to show the promotion of depletion of VTC complex proteins in phagophore entry into vacuoles. We found that Vtc4 depletion also promoted the entry of phagophores in *snf7*∆ cells into vacuoles as it did for *vps21*∆ cells, although it is more obvious in *vps21*∆ cells. These results revalidate the role of VTC complex proteins in negatively regulating phagophore entry into vacuoles and may also indicate a widespread role of the VTC complex in negatively regulating the entry of other cargos into yeast vacuoles.

## 2. Materials and Methods

### 2.1. Strains and Reagents

The *Saccharomyces cerevisiae* yeast strains used in this study are listed in Appendix A [9,15,25]. All yeast and *Escherichia coli* transformations were performed as previously described [26].

The parent *S. cerevisiae* yeast strain SEY6210 was used for fluorescence tagging and gene deletion to generate strains for this study. GFP-Atg8 expressed from the *pP_1K_Green fluorescent protein (GFP)-ATG8(406)* was constructed with 990 base pairs of *ATG8* 5′ sequence in front of the GFP-Atg8 open reading frame and linearized with EcoRV and integrated into the target strains [27]. GFP-Atg8- or mCherry-Atg8-tagged strains were confirmed for fluorescence phenotypes, and Vph1-GFP strains were further tagged with a *HIS3* cassette for mCherry-Atg8-tagged strains. A drug-resistance cassette (*hphMX4* or *kanMX3*) was used to replace the targeted gene in different background strains to generate deletion strains. Either Vph1-GFP tagging or gene deletion was achieved by using polymerase chain reaction (PCR) amplification with corresponding primers in Appendix A to produce DNA fragments, in combination with DNA fragment transformation for endogenous recombination. Vph1-GFP-tagged strains were selected on SD-His plates, and gene deletion strains were selected on YPD+Hygromycin B (H8080, Solarbio, Beijing, China) or YPD+Geneticin (G8160, Solarbio), depending on the drug-resistance cassette *hphMX4* or *kanMX3*, respectively. The deletion strains were further confirmed with diagnostic PCR with corresponding primers in Appendix A. The function of Vtc4 depletion in promoting the entry of phagophores into vacuoles in Vps21 or Snf7 depletion strain was examined with fluorescence microscopy observations and immunoblotting assays after the cells were grown under normal growth conditions or induced for autophagy.

Antibodies used in this study were mouse anti-GFP (sc-9996; Santa Cruz Biotechnology, Dallas, TX, USA), rabbit anti-glucose-6-phosphate dehydrogenase (G6PDH, A9521; Sigma-Aldrich, St. Louis, MO, USA), horseradish peroxidase (HRP)-linked goat anti-mouse Immunoglobulin G (IgG) (GAM007; Multisciences, Hangzhou, China), and HRP-linked goat anti-rabbit IgG (GAR007; Multisciences). All other chemical reagents used in this study were purchased from Sigma (St Louis, MO, USA) or from Amersco (Fair Lawn, NJ, USA), except for SynaptoRed, also known as FM4-64 from Biotium (70021; Hayward, CA, USA), and PCR polymerases and buffers from Takara Biotechnology (Dalian, China).

### 2.2. Yeast Growth, Induction of Autophagy and Microscopy Observation

Strains were grown on rich yeast extract peptone dextrose (YPD) plates or in YPD liquid at 26 °C for 3 days. If the strains were deletion strains, an additional 0.02% Geneticin or 0.03% Hygromycin B was added into medium. YPD liquid contains 2% (*w*/*v*) peptone, 1% yeast extract, 2% glucose, with an additional 2% agar for plate. The overnight culture was consecutively re-inoculated for three times at an initial optical density at 600 nm (OD_600_) of approximately 0.005–0.015 in 3 mL of YPD medium with rotation at 200 rpm to reach the mid-log phase for live-cell fluorescence microscopy.

Autophagy was induced by incubating cells in a synthetic liquid medium without nitrogen (SD-N). This medium comprised 0.17% yeast nitrogen base without amino acids and ammonium sulfate and 2% glucose. Briefly, cells were collected at the mid-log phase by centrifugation at 4000 rpm for 3 min. The supernatant was removed to add an equal volume of sterilized water to resuspend and wash away the nutrient from the pelleted cells for three repeats. The final supernatant was removed after centrifugation and replaced with a 60% volume of SD-N medium for starvation. The new culture was incubated for the indicated periods to observe fluorescence images and/or perform immunoblotting assays.

A final concentration of 1.6 μM FM4-64 was added in the indicated culture, and the culture was incubated at 26 °C for 2 h to stain the cell vacuoles before collecting the cells. The cells were examined on slides using a Leica (Wetzlar, Germany) laser scanning confocal microscope (TCS SP8), equipped with a high-contrast Plan Apochromat 100 × 1.4-NA oil-immersion objective lens, photomultiplier tube detectors and Hybrid detectors, and acquisition software Leica LAS X. More than three fields per sample were assessed. Those fluorescence images presented along with differential interference contrast (DIC) images were acquired using a Leica confocal microscope with LAS X program for single slices, and a central slice was selected. Images were processed for brightness and contrast with Photoshop CS6 (Adobe, San Jose, CA, USA) and assembled with Illustrator CS5 (Adobe) for publication [12].

### 2.3. Immunoblotting Analysis

Crude lysates from cells expressing GFP-Atg8 cells were subjected to immunoblotting assay to analyze autophagy processing in at least two independent experiments, as previously described [28,29]. Blots were probed with an anti-GFP antibody to determine the levels of GFP-Atg8 and GFP. An anti-G6PDH antibody was used to determine the expression levels of glucose-6-phosphate dehydrogenase (G6PDH) to display the loading situation. Immunoblots performed with indicated antibodies were subjected to electrochemiluminescence (ECL) HRP substrate (P90720; Millipore, Burlington, MA, USA) and exposed to X-ray film (XBT; Carestream, Rochester, NY, USA) to generate an image with an HQ-350XT film developer (Huqiu Imaging Technology, Suzhou, China). Immunoblotting bands were quantified using the ImageJ (National Institutes of Health, Bethesda, MD, USA) for band density in cases where the quantification was necessary. The relative percentage of GFP accounted for the total amount of GFP-Atg8, and GFP [(% GFP = (band density of GFP)/(band density of GFP-Atg8 + band density of GFP) × 100%] was used to indicate the autophagy level. Therefore, there is no need to normalize GFP-Atg8 and GFP bands to G6PDH loading band for each sample.

### 2.4. Statistical Analyses

The data are reported as the mean ± standard deviation (SD) calculated from at least two independent experiments. Student’s *t*-test was used to determine statistical significance. The *p*-value was used to demonstrate the significance, which was represented as: n.s., not significant; and **, *p* < 0.01.

## 3. Results

### 3.1. Vtc4 Depletion Promotes the Entry of GFP-Atg8 Clusters into FM4-64-Labeled Vacuoles in snf7∆ Cells

The VTC complex proteins were recently reported to negatively regulate the entry of misfolded glycosylphosphatidylinositol (GPI)-anchored proteins caused by Pep4 depletion into vacuoles with microautophagy [24]. In yeast, phagophores (unclosed autophagosomes) are closed by the ESCRT complex in a Vps21/Rab5-dependent manner [12]. Phagophores accumulate at vacuolar membranes in *vps21*∆ and ESCRT mutant cells under a short duration of autophagy induction with nitrogen starvation [12,13,14]. The absence of VTC complex proteins (either Vtc1, 2, 4, or 5) promotes the entry of GFP-Atg8-labeled phagophores into FM4-64-labeled vacuoles in *vps21*∆ cells, while the absence of VTC complex proteins itself did not obviously delay or promote these processes [15]. Snf7 is a core subunit of ESCRT subcomplex III (ESCRT-III), and Vps4 ATPase is an accessory protein for ESCRT. The absence of Snf7 or Vps4 results in the strongest autophagy defects in yeast ESCRT mutant cells [12]. To examine whether the absence of VTC complex proteins also promotes the entry of phagophores into vacuoles in ESCRT mutant cells, we chose *snf7*∆ cells as an example for ESCRT mutant cells. Before deleting *VTC4*, the phenotype of GFP-Atg8-labeled phagophore accumulation was confirmed with nitrogen starvation for 2 or 3 h in *snf7*∆ cells. This phenotype was consistent to that in the other ESCRT mutant *vps4*∆ cells (Appendix A), indicating the representativeness of *snf7*∆ cells for ESCRT mutant cells. We then deleted *VTC4* from *snf7*∆ cells to obtain *snf7*∆*vtc4*∆ cells. We examined the strains (*snf7*∆ and *snf7*∆*vtc4*∆) for phenotypes with the previous studied strains *vps21*∆ and *vps21*∆*vtc4*∆ [15] as controls besides the regular control strains (WT, *vtc4*∆ and *atg1*∆). Atg1 is a core protein for the autophagy machinery, and its absence inhibits the initiation of autophagy [30]. During autophagy in WT yeast cells, GFP-Atg8-labelled autophagosomes are delivered to vacuoles, and this process is inhibited in autophagy defective mutants [31]. With the increase in nitrogen starvation durations, GFP-Atg8 entered FM4-64-stained vacuoles in WT and *vtc4*∆ cells, but GFP-Atg8 was always in cytosol with an occasional dot on the vacuolar membrane in *atg1*∆ cells (Figure 1A). These expected results indicate that GFP-Atg8 was normally delivered to vacuoles in WT and *vtc4*∆ cells, but the delivery process was completely inhibited in *atg1*∆ cells, so that the experimental system is reliable. GFP-Atg8 clusters (phagophores) appeared in *vps21*∆ and *snf7*∆ cells between 2-8 h with nitrogen starvation. These phenotypes were more obvious between 2-4 h because phagophores gradually entered vacuoles with prolonged nitrogen starvation ((Figure 1B) and [15]). There were less GFP-Atg8 clusters on vacuolar membranes and more GFP-Atg8 signals insides vacuoles in *vps21*∆*vtc4*∆ and *snf7*∆*vtc4*∆ cells than those in *vps21*∆ and *snf7*∆ cells, respectively, under the same nitrogen starvation duration (Figure 1A,B). Quantifications of percentage (%) of cells with GFP-Atg8 clusters on vacuolar membranes support the conclusion from direct fluorescence observation (Figure 1C,D). Therefore, these results suggest that Vtc4 depletion promotes the entry of GFP-Atg8 clusters into FM4-64-labeled vacuoles in *snf7*∆ cells, consistent to that in *vps21*∆ cells.

### 3.2. Vtc4 Depletion Promotes GFP-Atg8 Degradation in Mean Value with no Significance in snf7∆ Cells

The increased entry of phagophores into vacuoles in *vps21*∆ cells with Vtc4 depletion promoted the degradation of GFP-Atg8 to become more stable GFP, only reflected in mean value with no significance [15]. To examine whether the increased entry of phagophores into vacuoles in *snf7*∆ cells with Vtc4 depletion also promoted GFP-Atg8 degradation, we examined the strains in Figure 1A for GFP-Atg8 degradation after 0, 2 and 8 h of nitrogen starvation. There was no obvious degradation of GFP-Atg8 in all strains at 0 h of nitrogen starvation. Massive GFP-Atg8 was degraded in WT and *vtc4*∆ cells at 2 h of nitrogen starvation, and this trend continued at 8 h of nitrogen starvation (Figure 2A). There was no obvious GFP-Atg8 degradation in *vps21*∆ and *vps21*∆*vtc4*∆ cells at 2 h of nitrogen starvation, although partial GFP-Atg8 degradation was observed in *snf7*∆ and *snf7*∆*vtc4*∆ cells at 2 h of nitrogen starvation. GFP-Atg8 degradation was observed in both *vps21*∆ and *snf7*∆ cells with or without Vtc4 depletion at 8 h of nitrogen starvation. Most importantly, there was a trace amount of increase in GFP-Atg8 degradation in *vps21*∆ and *snf7*∆ cells with Vtc4 depletion at 8 h of nitrogen starvation compared to those in *vps21*∆ and *snf7*∆ cells. No GFP-Atg8 degradation was observed in *atg1*∆ cells with all tested nitrogen starvation durations (Figure 2A). The degradation of GFP-Atg8 in *vps21*∆ and *snf7*∆ cells is highly dependent on the physiological condition, including cell density values (Figure 1D in [15]). Furthermore, the amount of GFP-Atg8 at the start of nitrogen starvation is different among strains not only in this study (Figure 2A) but also in another study [29]. The amount of GFP-Atg8 could increase or decrease in different mutant cells. We do not know the reason behind this yet, although the related mutation genes playing roles in regulation of transcription for expression or post-translation for degradation cannot be excluded. Therefore, we should not compare the absolute density of GFP bands. Instead, we always compared the relatively degraded GFP from GFP-Atg8 under the same growth conditions among different strains to represent the autophagy level. Through quantifying the amount of band densities of GFP and GFP-Atg8, the autophagy level can then be expressed with the percentage of GFP accounting for GFP and GFP-Atg8. The quantified percentage of GFP (%GFP) showed that the promotion of Vtc4 depletion on GFP-Atg8 degradation in *vps21*∆ and *snf7*∆ cells existed in mean values but with no significance (Figure 2B). These results suggest that Vtc4 depletion promotes the entry of phagophores into vacuoles but does not significantly increase GFP-Atg8 degradation in *snf7*∆ cells, consistent to that in *vps21*∆ cells.

### 3.3. Vtc4 Depletion Promotes the Entry of mCherry-Atg8 Clusters into Vph1-GFP-Labeled Vacuoles in snf7∆ Cells

FM4-64 stains both vacuolar membranes and phagophores (Figure 1A and [9]). As phagophores often accumulated next to the vacuolar membranes as clusters, it was difficult to distinguish their edges and conclude whether the phagophores entered vacuoles by red fluorescence. To examine the entry of phagophores into vacuoles more clearly, we labeled a vacuolar membrane protein Vph1 in mCherry-Atg8 strains with GFP at its C-terminal to show the vacuoles and to obtain mCherry-Atg8 Vph1-GFP strains. We confirmed that mCherry-Atg8 clusters accumulated on Vph1-GFP-labeled vacuolar membranes in *vps21*∆ and *snf7*∆ cells at 2 h of nitrogen starvation, although Vph1-GFP also marked the class E structures [32,33] in *snf7*∆ cells, and there were more cells with mCherry-Atg8 clusters in *vps21*∆ cells than in *snf7*∆ cells (Figure 3 and Appendix A). *VTC4* was deleted from mCherry-Atg8 Vph1-GFP-labeled *vps21*∆ and *snf7*∆ cells to obtain double mutant strains. The indicated strains in Figure 3A were grown and examined as in Figure 1A for red and green fluorescence. Without nitrogen starvation, mCherry-Atg8 dots were occasionally observed in WT and *vtc4*∆ cells, but almost no mCherry-Atg8 dots were observed in *vps21*∆, *vps21*∆*vtc4*∆ and *atg1*∆ cells. However, mCherry-Atg8 dots accumulated at the class E structures displayed with Vph1-GFP in *snf7*∆ and *snf7*∆*vtc4*∆ cells. When *vps21*∆ and *snf7*∆ cells were treated with nitrogen starvation for 2 h, accumulated mCherry-Atg8 clusters on vacuolar membranes in them become remarkable. However, such accumulated mCherry-Atg8 clusters decreased or disappeared when *VTC4* was deleted from *vps21*∆ and *snf7*∆ cells. In contrast, red signals inside vacuoles in *vps21*∆*vtc4*∆ and *snf7*∆*vtc4*∆ cells increased. When the cells were starved in SD-N for 8 h, red signals (some were mCherry-Atg8 clusters or puncta) could easily be found in *vps21*∆ and *snf7*∆ cells with or without Vtc4 depletion (Figure 3A). Quantifications of the percentage (%) of cells with mCherry-Atg8 clusters on vacuolar membranes support these direct fluorescence observations (Figure 3B). These results further confirmed that Vtc4 depletion promotes the entry of phagophores (mCherry-Atg8 clusters) into vacuoles in *snf7*∆ and *vps21*∆ cells.

## 4. Discussion

The entry of cargoes into vacuoles is regulated by multiple pathways and regulators in yeast cells. The fusion step is essential for cargoes in membrane structures to enter vacuoles. There are many positive regulators for vesicle–vacuole fusions in the endocytosis pathway or autophagosome–vacuole fusions in autophagy, such as a yeast Ypt7 module, including GEF subunits, Mon1 and Ccz1; Rab GTPase, Ypt7; tether, HOPS complex; SNARE protein, Vam3 [9]. Previously, only closed autophagosomes were supposed to fuse with vacuoles/lysosomes to be degraded [16]. In two-dimensional topology thinking, the double-membrane phagophores were thought to be unable to fuse with lysosomes/vacuoles to degrade enclosed cargoes. After the fusion of the outer membrane of the phagophore with the vacuolar membrane, it was thought that the cargoes inside phagophores would be expelled out. However, the autophagosome–vacuole fusion is a three-dimensional process, and the supposed expelling process will not occur autonomously [4]. Recently, phagophores were found to fuse with lysosomes in mammalian cells or with vacuoles in yeast cells [4,15,17]. The phagophores in vacuoles in *vps21*∆ and ESCRT mutant cells clearly indicate that phagophores enter vacuoles [15]. The detailed mechanisms for these fusion processes are still unknown; however, these three-dimensional fusion processes should not be denied with incorrect two-dimensional topology thinking [4]. Ypt7 and Vam3 are required for the fusion of phagophores with vacuoles [15]. Most likely, the other components of the Ypt7 module would also positively regulate this fusion process.

Negative regulators during fusion of stuffs to vacuoles were seldom reported, but such regulators do exist. The VTC complex proteins negatively regulate the entry of misfolded GPI-anchored proteins in *pep4*∆ cells into vacuoles during microautophagy [24]. The VTC complex proteins also negatively regulate the entry of phagophores into vacuoles in *vps21*∆ cells as the further depletion of either four out of five proteins in the VTC complex promoted the entry of phagophores into vacuoles and increased phagophore degradation [15]. If the depletion of the VTC complex protein also promoted the phagophores in ESCRT mutant *snf7*∆ cells to enter vacuoles under nitrogen starvation, we expected that there would be more phagophores inside vacuoles in *snf7*∆*vtc4*∆ cells than in *snf7*∆ cells. We compared phagophore localization and degradation in *snf7*∆ and *snf7*∆*vtc4*∆ cells in this study. The *vps21*∆ and *vps21*∆*vtc4*∆ cells were included as control cells. Our results showed that the percentage of cells with accumulated phagophores on vacuolar membranes in *vps21*∆ and *snf7*∆ cells significantly decreased when Vtc4 was depleted, no matter whether GFP-Atg8 or mCherry-Atg8 were used to label the phagophores (Figure 1 and Figure 3). The mean percentage of GFP degraded from GFP-Atg8 in *vps21*∆*vtc4*∆ and *snf7*∆*vtc4*∆ cells was always higher than that in *vps21*∆ and *snf7*∆ cells although without significance (Figure 2), which may be due to the relatively low GFP-Atg8 degradation in mutants due to a possible decrease in hydrolysis ability. Alternatively, the decreased degradation of VTC complex proteins through vacuolar proteases in ESCRT mutant cells could not be excluded, as some Vtc5 is rerouted in the vacuolar lumen by ESCRT and immediately undergoes protease degradation [34]. If the degradation of VTC complex proteins in *snf7*∆ cells was decreased, there might be more VTC complex proteins to inhibit the entry of phagophores into vacuoles and be degraded. In addition, the degradation of GFP-Atg8 is highly dependent on the cell density, availability of vacuolar hydrolases and the nitrogen starvation time in *vps21*∆ and ESCRT mutant cells (Figures 1C–F, S1C and S6B,D in [15]). We do not know how these mutations affect the dynamic ability of hydrolases or the degradation of VTC complex proteins through vacuolar proteases. We always control the experiment conditions to be the same, with the strain as the only variation in the same experiment. Under this situation, we can strictly compare the difference in GFP-Atg8 degradation between *vps21*∆ and *vps21*∆*vtc4*∆ cells or *snf7*∆ and *snf7*∆*vtc4*∆ cells at the same nitrogen starvation time, as shown in Figure 2. However, when the changes in GFP-Atg8 degradation are minor and the sensitivity of ImageJ measurement always has limitations, the difference in the calculated percentage of GFP may be not significant between *vps21*∆ and *vps21*∆*vtc4*∆ cells or *snf7*∆ and *snf7*∆*vtc4*∆ cells. All these data are not related to florescence degradation or excessive exposure of the film. Finally, the molecular mechanism of Vtc4 depletion in promoting phagophore entry into vacuoles is still unclear, which needs further investigation.

ESCRT complex is a multiple subunit complex. ESCRT-III subunit Snf7 and AAA-ATPase Vps4 are the most frequent and well-studied subunits for their roles in vesicle trafficking and autophagy [35]. In our previous study in [12], we already showed that most ESCRT mutant cells, especially *snf7*∆ and *vps4*∆ cells, had similar and the strongest phenotypes for phagophore accumulation. We confirmed that the phagophore accumulation phenotypes in *snf7*∆ and *vps4*∆ cells are quite similar either under short or prolonged periods of nitrogen starvation (Appendix A and [12,15,36]). Therefore, the study contents for *snf7*∆ cells in this study are most likely representative for ESCRT mutant cells. Besides Vtc4, depletion of Vtc1, 2 or 5 also promoted the entry of phagophores into vacuoles in *vps21*∆ cells at similar levels [15]. The examinations of Vtc4 depletion in *snf7*∆ cells most likely will be representative for the examinations of the depletion of any VTC complex proteins (Vtc1, 2, 4 and 5) in ESCRT mutant cells. We did not examine the effects of depletion of other VTC complex proteins in *snf7*∆ cells, and we also did not examine the depletion of any VTC complex proteins (Vtc1, 2, 4 and 5) in *vps4*∆ cells. If these experiments were performed, it is expected that they will generate similar results and conclusions about Vtc4 depletion in *snf7*∆ cells, although this needs future experiments to support it. Mammalian ESCRT complex subunits are also required for phagophore closure, as phagophores accumulate in cytosol when the key ESCRT complex subunits are absent [36]. The size of the mammalian phagophore or autophagosome is relatively bigger than the size of the lysosome. There is no chance for mammalian phagophores or autophagosomes to enter lysosomes. In contrast, multiple mammalian lysosomes fuse with a phagophore or autophagosome to provide hydrolases to degrade them for recycling [37]. The similar positive fusion regulators as in yeast cells for phagophore or autophagosome–vacuole fusion are required for autophagosome–lysosome fusion in mammalian cells [38]. However, it is unclear whether there are negative regulators like VTC complex proteins in mammalian cells to promote the phagophore–lysosome fusion.

Our previous study showed that the entry of phagophores in *vps21*∆ cells was promoted by the depletion of VTC complex proteins [15]. This study further confirmed and broadened the promotion effect of depletion of the VTC complex protein on phagophore entry into vacuoles in yeast by using ESCRT mutant. These findings not only expand the functions of VTC complex but also provide information for the regulation of phagophores into vacuoles. Vtc4 depletion promotes the entry of phagophores into vacuoles in *vps21*∆ or *snf7*∆ cells; however, it is unclear whether Vps21 or Snf7 and Vtc4 act in a dependent way. Most likely, the accumulation of phagophores outside vacuoles by Vps21 or Snf7 depletion under nitrogen starvation provides a convenient way to study the effect of Vtc4 depletion to promote the phagophore entry. Vtc4 depletion did not have obvious promotion effect if the core four strains in Figure 1B were not starved to induce autophagy to form phagophores. We also noticed that the amount of GFP-Atg8 degradation in *vtc4*∆ cells was also more than that in wild-type cells in the mean value in Figure 2. We do not think that Vps21 and Vtc4 or Snf7 and Vtc4 act in a dependent way. We proposed that the VTC complex may have more roles to negatively regulate other cargoes outside or on vacuolar membranes to enter vacuoles in yeast. However, it is unclear whether such a mechanism also exists in mammalian cells.

## Figures and Tables

**Figure 1 jof-09-01003-f001:**
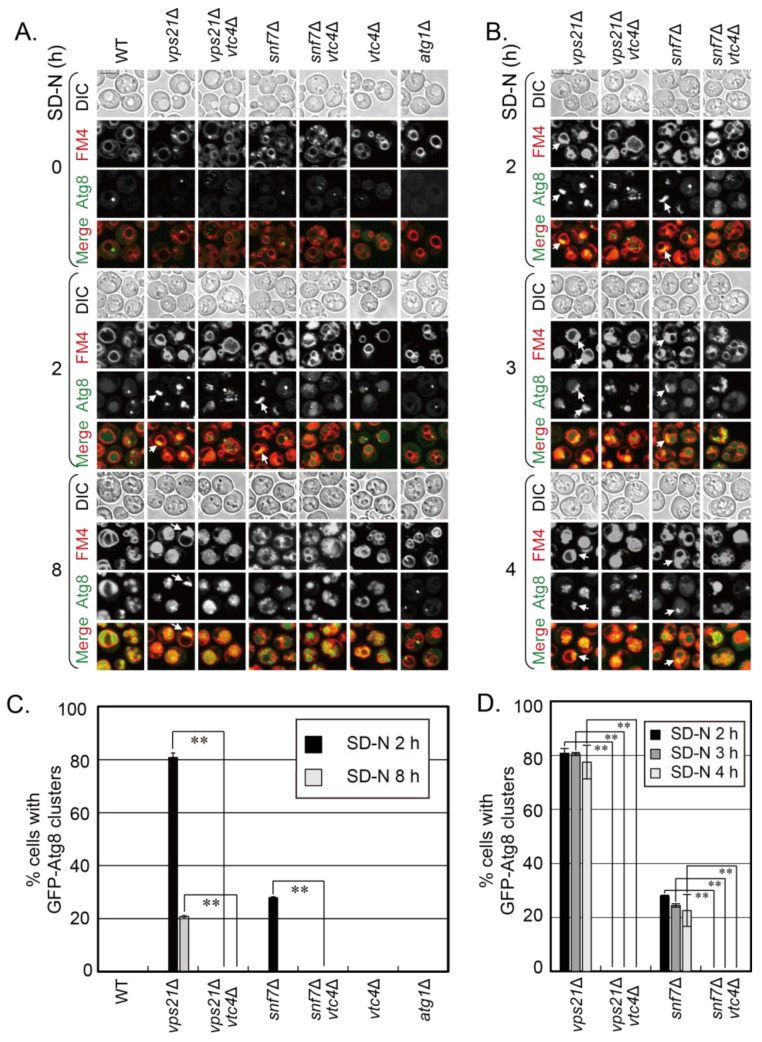
Vtc4 depletion promotes the entry of phagophores into vacuoles in *snf7*∆ cells under nitrogen starvation. (**A**) Vtc4 depletion promoted the entry of GFP-Atg8-labeled phagophores in *vps21*∆ and *snf7*∆ cells to FM4-64-stained vacuoles under nitrogen starvation (SD-N). The *vps21*∆ and *vps21*∆*vtc4*∆ cells served as control cells for studying Vtc4 depletion in promoting the entry of phagophores in *snf7*∆ cells into vacuoles. The indicated GFP-Atg8-labeled cells were grown and examined as previously described [15]. In brief, the cells were grown in rich medium (YPD) to mid-log phase and then shifted to SD-N medium for indicated durations to observe GFP-Atg8 localization and FM4-64 signals. (**B**) The indicated four core strains related to phagophore accumulation in SD-N medium between 3 and 4 h were further observed and presented. The data at 2 h reorganized from panel A were for comparison with the data at 3 and 4 h. Scales bars at the left top picture in the corresponding panel, 5 μm; arrows, phagophore clusters; DIC, differential interference contrast; FM4, FM4-64. (**C**,**D**) Quantification of the percentages (%) of cells with GFP-Atg8 clusters in the strains in panels A and B, representatively. Over 200 cells were counted for each strain. Statistical significance was analyzed using Student’s *t*-test (n ≥ 2 experiments). **, *p* < 0.01. The results shown represent two independent experiments.

**Figure 2 jof-09-01003-f002:**
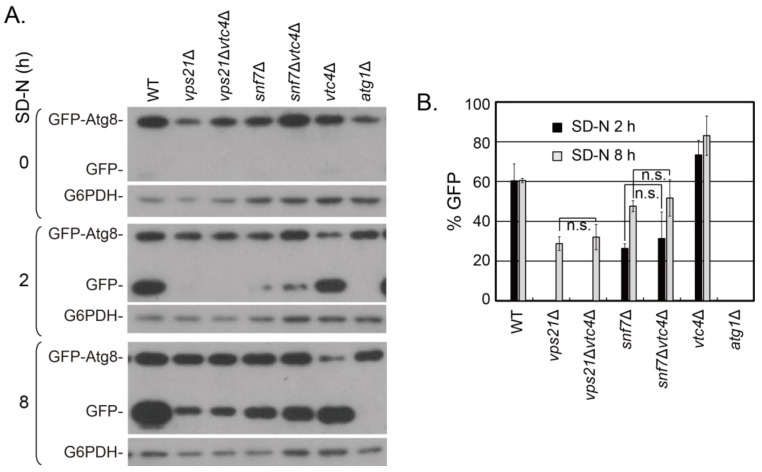
The bands and quantified mean value indicate that Vtc4 depletion promotes the degradation of GFP-Atg8 in *snf7*∆ cells under nitrogen starvation but with no significance. (**A**) Immunoblotting assays showed that Vtc4 depletion promoted the degradation of GFP-Atg8 in *vps21*∆ and *snf7*∆ cells under nitrogen starvation for 8 h. The *vps21*∆ and *vps21*∆*vtc4*∆ cells served as control cells for studying Vtc4 depletion in promoting the degradation of GFP-Atg8 to be more stable GFP. The indicated GFP-Atg8-labeled cells were grown and collected at indicated durations as in Figure 1A. The collected ~4 OD_600_ cells were lysed in 200 μL sample buffer to receive ~100 μL crude lysate to examine GFP-Atg8-labeled phagophore degradation with 5–10 μL per lane, as previously described [15]. (**B**) The % GFP was quantified after 2 and 8 h of SD-N treatment for the samples presented in panel A, which had GFP-Atg8 degradation. We did not perform data normalization as we used relative percentage of degraded GFP amount to show the autophagy process, and data normalization will not change the percentage. The quantitative data are presented as the mean +/− SD. n.s., not significant. The results shown represent two independent experiments.

**Figure 3 jof-09-01003-f003:**
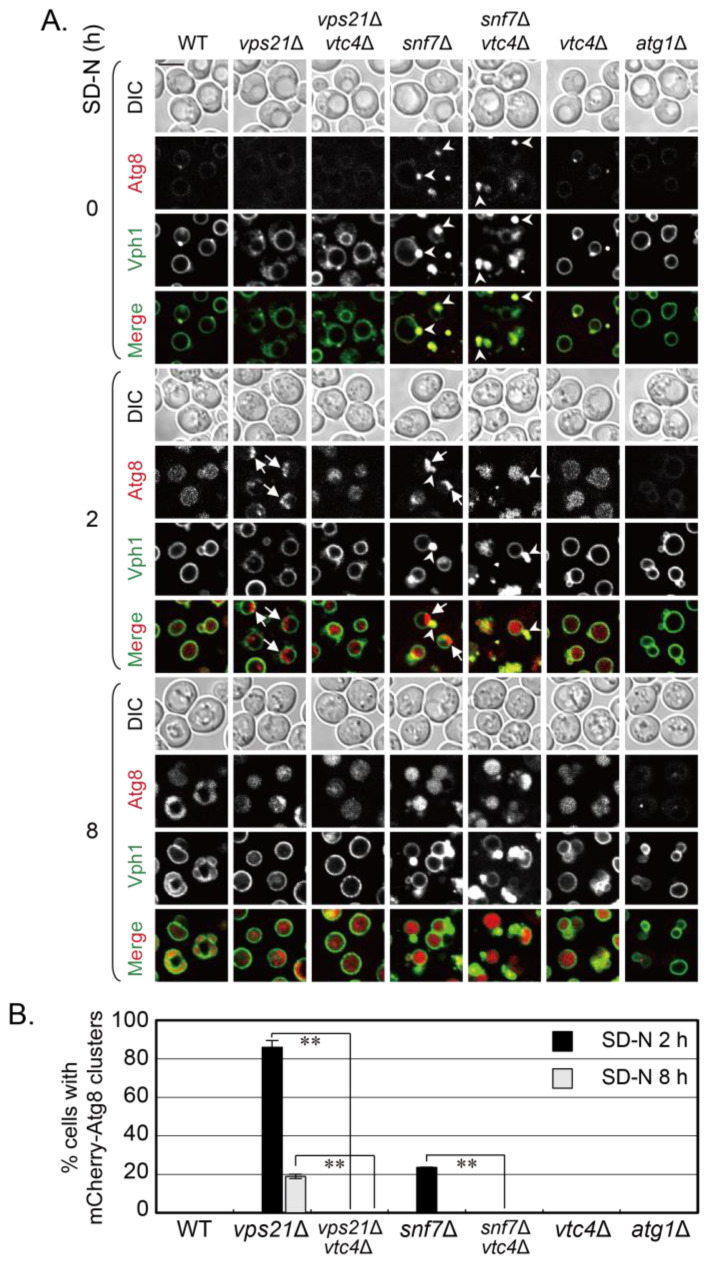
Using mCherry-Atg8-labeled phagophores to show that Vtc4 depletion promotes the entry of phagophores into Vph1-GFP-labeled vacuoles in *snf7*∆ cells under nitrogen starvation. (**A**) Vtc4 depletion promoted the entry of mCherry-Atg8-labeled phagophores into Vph1-GFP-labeled vacuoles in *vps21*∆ and *snf7*∆ cells under nitrogen starvation at 2 and 8 h. The *vps21*∆ and *vps21*∆*vtc4*∆ cells served as control cells for studying Vtc4 depletion in promoting the entry of phagophores into vacuoles in *snf7*∆ cells. The indicated mCherry-Atg8 Vph1-GFP-labeled cells were grown and examined as in Figure 1A. Scales bars at the left-top picture in panel A, 5 μm; arrows, mCherry-Atg8 clusters; arrowheads, class E structures; DIC, differential interference contrast. (**B**) Quantification of the percentages of cells with mCherry-Atg8 clusters in the strains represented in panel A. Over 200 cells were counted for each strain. **, *p* < 0.01. The results shown represent two independent experiments.

## Data Availability

Data supporting reported results are presented in this paper along with its Appendix A.

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
