# Peer review of "Vtc4 Promotes the Entry of Phagophores into Vacuoles in the Saccharomyces cerevisiae Snf7 Mutant Cell"

_jof, 2023, doi:10.3390/jof9101003_

Round 1

Reviewer 1 Report

The paper represents new and interesting part of ongoing research on endocytosis and autophagy.

20  Meaning of “staining circle” is not clear to me.

37-40 “Vesicles … are positively regulated …”. Maybe rephrase (fusion of vesicles is positively regulated).

48 “However, recently it is 48 clear…”.  Maybe: “However, recently it became clear…”.

82 “The deletion gene…” Maybe “The deleted gene …”.

79 There are no mutations listed for mel gene (which one of several?) in genotype of SEY6210 in initial paper (PMC365587).

86 “The function of Vtc4 in Vps21 or Snf7 depletion strain was examined with fluorescence 86 microscopy observations…” What function of the strain?

84-85 “Either Vph1-GFP tagging or gene deletion was achieved by using polymerase chain reaction (PCR) amplification and recombination techniques as previously described [11].” In the cited paper (PMC9562215) this procedure is described as “Deletion strains in the haploid state were obtained by replacing the complete open-reading frame of the targeted genes with amino acid or drug cassette encoding genes using PCR-based homologous recombination with strains developed in previous studies.” This is not informative. Provide primers (in supplementary materials) that were used to produce DNA fragments for transformations.

114-115 “A final concentration of 1.6 μM FM4‐64 (70021; Biotium) was added and incubated 114 for 2 h to stain vacuoles before collecting the cells.” Maybe: “… and cells were incubated …”.

125 “Crude lysates from cells expressing GFP-Atg8 cells…”

If possible, add results for strain without snf7∆ and vps4∆ as negative control in figure S1.

158 membrane in atg1∆ cells

204-207 As I understand the results of the statistical analysis presented in the Figure 2B demonstrate that there is no significant difference in GFP-Atg8 degradation between snf7∆ and snf7∆vtc4∆. Thus, it cannot be suggested that “Vtc4 depletion promotes GFP-Atg8 degradation in snf7∆ cells”. Moreover, no significant difference is observed for vps21∆ and vps21∆ vtc4∆ which contradicts previous results (186-187 “The facilitation of phagophores into vacuoles in vps21∆ cells with Vtc4 depletion 186 slightly promoted GFP-Atg8 degradation to become more stable GFP [11].”) This whole part (3.2.) and corresponding part of discussion should be reworked to correctly represent these results and found contradictions.

216 “mean +/- STD”. Maybe SD?

Figure 2 – It looks like the amount of GFP-Atg8 is different for different strains at the start of the nitrogen starvation. Can the described deletions influence the expression of GFP-Atg8 chimera gene? I tried to find the exact description of this GFP-Atg8 expression cassette in cited previous studies. But I was not able to get reliable understanding of its structure. Which promoter is used for expression of GFP-Atg8? Can its activity be changed by described deletions? Can this influence the results?

232 “but almost no mCherry-Atg8 dot in vps21∆, vps21∆vtc4∆ and atg1∆ cells.” Maybe “but almost no mCherry-Atg8 dots were observed  in vps21∆, vps21∆vtc4∆ and atg1∆ cells.” Overall, although I am not qualified to assess the quality of English in this paper I feel that extensive editing of English language is needed.

306 What is “(2018-NC)”? A citation? There is also no citation for [26]. All citations should be checked.

Author Response

The Review Report (Reviewer 1)

The paper represents new and interesting part of ongoing research on endocytosis and autophagy.

Thank you for spending time to review our manuscript and providing invaluable comments. We enhanced the reference citation and relevance, the description of detailed method, and the interpretation of results for the supposed conclusion. Your other comments were addressed point-by-point below.

Comment 1: 20  Meaning of “staining circle” is not clear to me.

Response 1: “FM4-64 staining circle or Vph1-GFP” was revised to “FM4-64 stained or Vph1-GFP labeled membrane structures” in line 23.

Comment 2: 37-40 “Vesicles … are positively regulated …”. Maybe rephrase (fusion of vesicles is positively regulated).

Response 2: “Vesicles from endocytosis and autophagosomes from autophagy fuse with lysosomes/vacuoles and are positively regulated by the fusion machinery” was changed to “Both vesicles from endocytosis and autophagosomes from autophagy can fuse with lysosomes/vacuoles. These fusion steps are positively regulated by the fusion machinery” in line 44-46.

Comment 3: 48 “However, recently it is clear…”.  Maybe: “However, recently it became clear…”.

Response 3: “is” was changed to “became” in line 58.

Comment 4: 82 “The deletion gene…” Maybe “The deleted gene …”.

Response 4: “The deletion gene in different background strains was replaced with a drug-resistance cassette (hphMX4 or kanMX3)” was changed to “A drug-resistance cassette (hphMX4 or kanMX3) was used to replace the targeted gene in different background strains to generate deletion strains” in line 112-114.

Comment 5: 79 There are no mutations listed for mel gene (which one of several?) in genotype of SEY6210 in initial paper (PMC365587).

Response 5: SEY6210 was obtained from another lab and some literatures listed mel gene for it (PMC2592676; PMC2366845; PMID: 11821410) and we checked the initial paper (PMC365587) and found mel gene was not listed for it. We sticked to the initial paper and just deleted mel for the genotype of SEY6210 as shown in Table 1.

Comment 6: 86 “The function of Vtc4 in Vps21 or Snf7 depletion strain was examined with fluorescence microscopy observations…” What function of the strain?

Response 6: “The function of Vtc4 in Vps21 or Snf7 depletion strain was examined with fluorescence microscopy observations…” was changed to “The function of Vtc4 depletion in promoting the entry of phagophores into vacuoles in Vps21 or Snf7 depletion strain was examined with fluorescence microscopy observations” in line 121-123.

Comments 7: 84-85 “Either Vph1-GFP tagging or gene deletion was achieved by using polymerase chain reaction (PCR) amplification and recombination techniques as previously described [11].” In the cited paper (PMC9562215) this procedure is described as “Deletion strains in the haploid state were obtained by replacing the complete open-reading frame of the targeted genes with amino acid or drug cassette encoding genes using PCR-based homologous recombination with strains developed in previous studies.” This is not informative. Provide primers (in supplementary materials) that were used to produce DNA fragments for transformations.

Response 7: Thank you for your suggestion. The primers were listed in Table 2 in line 125-126. In addition, the mentioned sentence “Either Vph1-GFP tagging or gene deletion was achieved by using polymerase chain reaction (PCR) amplification and recombination techniques as previously described [11].” was changed to “Either Vph1-GFP tagging or gene deletion was achieved by using polymerase chain reaction (PCR) amplification with corresponding primers in Table 2 to produce DNA fragments, in combination with DNA fragment transformation for endogenous recombination. Vph1-GFP tagged strains were selected on SD-His plates, and gene deletion strains were selected on YPD+Hygromycin B (H8080, Solarbio) or YPD+Geneticin (G8160, Solarbio) depending on the drug-resistance cassette hphMX4 or kanMX3, respectively. The deletion strains were further confirmed with diagnostic PCR with corresponding primers of Table 2” to provide more details in line 114-121.

Comment 8: 114-115 “A final concentration of 1.6 μM FM4‐64 (70021; Biotium) was added and incubated for 2 h to stain vacuoles before collecting the cells.” Maybe: “… and cells were incubated …”.

Response 8: The mentioned sentence was changed to “A final concentration of 1.6 μM FM4‐64 was added in the indicated culture and the culture was incubated at 26°C for 2 h to stain the cell vacuoles before collecting the cells” in line 152-153.

Comments 9: 125 “Crude lysates from cells expressing GFP-Atg8 cells…”

If possible, add results for strain without snf7∆ and vps4∆ as negative control in figure S1.

Response 9: The strain without snf7∆ and vps4∆ is wild-type (WT). We added the data of “WT” starved in SD-N for 2 h in Figure S1 and updated the figure legend in line 440-442.

Comments 10: 158 membrane in atg1∆ cells

Response 10: Thanks. Corrected in line 210.

Comments 11: 204-207 As I understand the results of the statistical analysis presented in the Figure 2B demonstrate that there is no significant difference in GFP-Atg8 degradation between snf7∆ and snf7∆vtc4∆. Thus, it cannot be suggested that “Vtc4 depletion promotes GFP-Atg8 degradation in snf7∆ cells”. Moreover, no significant difference is observed for vps21∆ and vps21∆ vtc4∆ which contradicts previous results (186-187 “The facilitation of phagophores into vacuoles in vps21∆ cells with Vtc4 depletion slightly promoted GFP-Atg8 degradation to become more stable GFP [11].”) This whole part (3.2.) and corresponding part of discussion should be reworked to correctly represent these results and found contradictions.

Response 11: Thank you for your comments. Yes, there is no significant difference for GFP-Atg8 degradation between snf7∆ and snf7vtc4∆ cells, or between vps21∆ and vps21vtc4∆ cells in this study. There is also no significant difference for GFP-Atg8 degradation between vps21∆ and vps21vtc4∆ cells in our previous study (Figure S3B-C in [11]). Therefore, there is no contradiction in results. We agreed that we should not conclude that Vtc4 depletion promotes GFP-Atg8 degradation in snf7∆ cells or vps21∆ cells as there is no significant difference in quantification.

However, no matter from the GFP band morphology or quantification of % GFP for GFP-Atg8 degradation in the above studies, we always found the GFP band is a little bit wide or black in double mutants (snf7vtc4∆ and vps21vtc4∆) than in their corresponding single mutants (snf7∆ and vps21∆), % GFP is higher in double mutants (snf7vtc4∆ and vps21vtc4∆) than in their corresponding single mutants (snf7∆ and vps21∆) in mean values.

We revised the related sentences to be “The increased entry of phagophores into vacuoles in vps21∆ cells with Vtc4 depletion promoted the degradation of GFP-Atg8 to become more stable GFP, only reflected in mean value with no significance” in line 242-244, and to be “These results suggest that Vtc4 depletion promotes the entry of phagophores into vacuoles but does not significantly increase GFP-Atg8 degradation in snf7∆ cells, consistent to that in vps21∆ cells.” in line 271-273.

We provided explanations for the variations of band densities of initial GFP-Atg8 and why we chose to compare the percentage of GFP for GFP-Atg8 degradation in line 257-268. Furthermore, we discussed the possible reasons for the observed difference but with no significance in line 364-384.

Comment 12: 216 “mean +/- STD”. Maybe SD?

Response 12: Thanks. Corrected in line 287.

Comments 13: Figure 2 – It looks like the amount of GFP-Atg8 is different for different strains at the start of the nitrogen starvation. Can the described deletions influence the expression of GFP-Atg8 chimera gene? I tried to find the exact description of this GFP-Atg8 expression cassette in cited previous studies. But I was not able to get reliable understanding of its structure. Which promoter is used for expression of GFP-Atg8? Can its activity be changed by described deletions? Can this influence the results?

Response 13: We noticed that the amount of GFP-Atg8 is different for different strains at the start of the nitrogen starvation not only in this study but also in another study (PMC3538905). The amount of GFP-Atg8 can increase or decrease in mutant cells. We don’t know the reason yet, although the related mutation genes playing roles in regulation on transcription for expression or post-translation for degradation cannot be excluded. Therefore, we should not compare the absolute density of GFP bands. Instead of, we always compared the relatively degraded GFP from GFP-Atg8 under the same growth condition among different strains to represent the autophagy level. We don’t think the different amount of GFP-Atg8 expression will influence the results much. The relative percentage of GFP accounted for the total amount of GFP-Atg8 and GFP [(% GFP = (band density of GFP) / (band density of GFP-Atg8 + band density of GFP) ´ 100%] was used to indicate the autophagy level. Again, these concerns were addressed in line 257-268 and discussed in line 364-384.

The exact description of this GFP-Atg8 expression cassette can be found in a previous study (PMC2488302), named pP1KGreen fluorescent protein (GFP)-ATG8(406). Plasmid pP1KGreen fluorescent protein (GFP)-ATG8(406) contains 990 base pairs of ATG8 5′sequence in front of the GFP-Atg8 open reading frame. Therefore, it used its own promoter. The Atg8-expressing plasmid was linearized with EcoRV and integrated into the target strains. Please refer to line 107-110.

Comments 14: 232 “but almost no mCherry-Atg8 dot in vps21∆, vps21∆vtc4∆ and atg1∆ cells.” Maybe “but almost no mCherry-Atg8 dots were observed in vps21∆, vps21∆vtc4∆ and atg1∆ cells.”

Response 14: Thanks. Corrected in line 303-304.

Comments 15: Overall, although I am not qualified to assess the quality of English in this paper, I feel that extensive editing of English language is needed.

Response 15: We tried our best to improve the language. All the changes were marked in blue in the revision texts. If you still find big issues for language, please feel free to let us know. We will further revise in the next round of revision.

Comments 16: 306 What is “(2018-NC)”? A citation? There is also no citation for [26]. All citations should be checked.

Response 16: Thanks. 2018-NC is a label for inserting this reference “Takahashi, Y.; He, H.; Tang, Z.; Hattori, T.; Liu, Y.; Young, M.M.; Serfass, J.M.; Chen, L.; Gebru, M.; Chen, C.; et al. An autophagy assay reveals the ESCRT-III component CHMP2A as a regulator of phagophore closure. Nat Commun 2018, 9, 2855, doi:10.1038/s41467-018-05254-w.” We are sorry for forgetting to do that. Corrected to be the new reference [36] now.

[26] should be the new reference [38]. We are sorry for this mistake. We switched Endnote in different computers and caused some incompatibility but we did not notice them before the initial submission.

All citations in the revision were checked now.

Reviewer 2 Report

The manuscript entitled “Vtc4 promotes the entry of phagophores into vacuoles in Snf7 mutant yeasts” by Chen & Liang attempt to identify the role of a subunit of VTC complex during autophagy. However, the authors have not highlighted the need for Snf17 deleted background for this understanding. This study is also a replica of methods used in their previous publication in PLoS Genet 18(10): e1010431 published in 2022 (Ref 11) and provides additional observations to the functional roles of vtc subunits. Therefore, the basis of the result interpretation has no major advances or novelty as it explains in its current format. However, the results are interesting and would be useful to further explore the molecular mechanisms of autophagy.

There are no major concerns on the research methodology or observation although the authors are required to explain further to improve the clarity and rationale for this study and how it deviates from reference #11. Below are the major comments for each section.

This study used the model yeast, S. cerevisiae and authors should mention the yeast species as S. cerevisiae is not the only model yeasts in modern mycology. Could be included in the title, in the Table title or methods etc.

Introduction: As of now, the introduction lacks cohesiveness and flow. Authors should improve the flow by explaining phagophore (line 34), Vps21 (line 43), link between ESCART and Snf7, macro- and micro-autophagy to give a more meaningful reader friendly background. It is a question that the authors provide sufficient background research by seeing the limited number of reference list. Further line 51-54, need references to support your ideas.

Line 54: why sporulation?

Authors should explain how this study is different from Ref 11 and what is the unique focus for this study. That means, the rationale for the research question is not well described. Needs to address the knowledge gap filled by this work.

Materials & Methods: Table 1 includes a figure column and authors should rethink the need for that. It may be better to replace the column with the abbreviated genotype strain name used in the figures rather than the figure numbers.

Line 78: Background strain should be replaced with Parent strain? Parent strain should be included in table 1.

Line 80: ‘current strains’ à to obtain strains generated for this study. However, line 78-80 could be brief out when the parent strain is included in the table.

Line 82-83: Confusing sentence. Include the gene deletion strategy instead.

Line 86: What do you mean by ‘function’ here?

Line 85 and 94: need rewriting with technical terms rather merely saying described elsewhere.

Line 97-102: General microbiology techniques can be removed.

Line 112: rewrite to improve clarity.

Section 2.3: Did you normalize your data based on the loading control? If so, explain in methods and Fig 2. Since the loading control is not homogenous as of now, indicate how much protein was added to each lane.

Results: The authors represented the results in a very neat way with a well detailed figure caption. However, the results section should be strengthened to support observations related to existing literature.

The authors should explain why a Snf7 deletion background was chosen. What do you mean by “autophagy is normal” in line 159? Also ‘slight’ in line 187 and 189? Remove or explain such vague statements with a proper quantitative or qualitative term/description. The authors should also explain how these single and double mutant strains contribute to the autophagy process or the rationale behind selecting these mutant types.

In Fig 1, authors also observe the four core strains in early time gaps. Why?

The scale bars on the microscopy images were almost invisible therefore write down the location in the figure caption to guide the reader.

Line 220: rewrite to improve clarity.

Line 228: Does Fig 1 also include data from mCherry tagged strains?

Line 229: What do you mean? Where?

Figure 3: Instead of “another way” use a meaningful title. For example, “mCherry labelled …..” etc.

Discussion: Many statements require citations.

Line 280-281: Rather than mentioning “as an example”, authors should explain the rationale behind using the double mutant.

Line 287: Explain what is meant by ‘more’ here comparatively.

Line 290: Is this due to the florescence degradation or excessive exposure of the film? Could it be the degradation of vtc protein through vacuolar proteases. It has been shown previously that some Vtc proteins are rerouted in the vacuolar lumen by EXCRT and immediately undergo protease degradation. Discuss this with relevant to previous literature on other subunits of VTC complex.

Line 300- 303: Authors should have a valid reason rather than a random pick for the selected mutants and explain why other proteins require further investigation.

Line 306: What is 2018-NC?

The authors should be able to discuss their findings in relation to past findings to provide the significance of this work.

If vtc4 depletion promotes phagophores entry, do you think that vps21 and vtc4 act in a dependent way? Explain.

Some corrections are required to improve the clarity and flow of the sentences.

Author Response

The Review Report (Reviewer 2)

Comments 1: The manuscript entitled “Vtc4 promotes the entry of phagophores into vacuoles in Snf7 mutant yeasts” by Chen & Liang attempt to identify the role of a subunit of VTC complex during autophagy. However, the authors have not highlighted the need for Snf7 deleted background for this understanding. This study is also a replica of methods used in their previous publication in PLoS Genet 18(10): e1010431 published in 2022 (Ref 11) and provides additional observations to the functional roles of vtc subunits. Therefore, the basis of the result interpretation has no major advances or novelty as it explains in its current format. However, the results are interesting and would be useful to further explore the molecular mechanisms of autophagy.

There are no major concerns on the research methodology or observation although the authors are required to explain further to improve the clarity and rationale for this study and how it deviates from reference #11.

Response 1: Thank you for spending time to review our manuscript and providing constructive comments. Yes, this study is an extension study of our previous study published in PLoS Genet 18(10): e1010431 in 2022 (Ref 11, [Ref 15 in revision]), technically with a replica of methods used in that paper.  In Ref 15, we found that the phagophores accumulated in both vps21∆ and ESCRT mutant cells (such as snf7∆ and vps4∆ cells) entered vacuoles under prolonged nitrogen starvation, and the depletion of VTC complex proteins (Vtc1, 2, 4, 5) promoted the entry of phagophores in vps21∆ into vacuoles. We did not study and did not know whether the depletion of VTC complex proteins (Vtc1, 2, 4, 5) promoted the entry of phagophores in ESCRT mutant cells into vacuoles in that paper. The main purpose of this study is to answer these questions. We used Vtc4 as a representative for VTC complex proteins and snf7∆ cells as a representative for ESCRT mutant cells to carry out this study. The new results may strengthen or deny a universal role of depletion of VTC complex protein for promoting the entry of phagophores into vacuoles, depending on the results from different strain resources. Therefore, we think this study is important for the field.

Below are the major comments for each section.

Comments 2: This study used the model yeast, S. cerevisiae and authors should mention the yeast species as S. cerevisiae is not the only model yeasts in modern mycology. Could be included in the title, in the Table title or methods etc.

Response 2: Thank you for your suggestion. We added the yeast species name S. cerevisiae in the title, in the Table title and method section.

Introduction:

Comments 3: As of now, the introduction lacks cohesiveness and flow. Authors should improve the flow by explaining phagophore (line 34), Vps21 (line 43), link between ESCRT and Snf7, macro- and micro-autophagy to give a more meaningful reader friendly background. It is a question that the authors provide sufficient background research by seeing the limited number of reference list. Further line 51-54, need references to support your ideas.

Response 3: All the mentioned terms were described in more details with reference supporting now in the introduction section. We revised the mentioned line 51-54 to the following information “Failure of autophagosome biogenesis and phagophore closure results in interrupted autophagy process, which compromises homeostasis and leads to various diseases, including metabolic disorders, neurodegeneration and cancer [18]. The fusion of phagophores with lysosomes/vacuoles partially restores cargo degradation and recycling, maintaining autophagy to a certain degree to resist stresses [4,15]” with references to support our claim in new line 60-65.

Comments 4: Line 54: why sporulation?

Response 4: Sporulation in diploid yeast cells needs autophagy and long incubation time of nitrogen starvation in the presence of a poor carbon source and sporulation was absent if autophagy was completely interrupted. The entry of phagophores in haploid or diploid vps21∆ cells into vacuoles occurred under prolonged nitrogen starvation. It is a good match to use sporulation to test whether the autophagy process in vps21∆/vps21∆ diploid cells was completely interrupted. Our results showed that the autophagy process in vps21∆/vps21∆ was impaired but not completely interrupted, not like atg1∆/atg1∆ and pep4∆/pep4∆ cells. We incorporated these background knowledges in line 65-72.

Comments 5: Authors should explain how this study is different from Ref 11 and what is the unique focus for this study. That means, the rationale for the research question is not well described. Needs to address the knowledge gap filled by this work.

Response 5: We addressed this concern in Response 1. Briefly, the previous Ref 11 (Ref 15 in revision) studied the promotion effect of the depletion of VTC complex proteins (Vtc1, 2, 4,5) on the entry of phagophores in the Rab5 GTPase Vps21 mutant cells. This study tried to examine whether the depletion of Vtc4 can also promote the entry of phagophores accumulated in ESCRT Snf7 mutant cells into vacuoles. In the end, we wanted to know whether the depletion of VTC complex protein can generally promote the entry of phagophores from different background strains into vacuoles. We paid more attention to the rationale of this study and revised the texts in the corresponding places. Please refer to line 78-99.

Materials & Methods:

Comments 6: Table 1 includes a figure column and authors should rethink the need for that. It may be better to replace the column with the abbreviated genotype strain name used in the figures rather than the figure numbers.

Response 6: Thank you for your comments, we added the abbreviated genotype strain name before the figure number. This way not only helps us and readers to understand the results, but also provides convenience for us to further use these strains or provide strains for other labs.

Comments 7: Line 78: Background strain should be replaced with Parent strain? Parent strain should be included in table 1.

Response 7: Corrected in line 106 with parent S. cerevisiae yeast strain SEY6210 and the parent strain was included in Table 1.

Comments 8: Line 80: ‘current strains’ à to obtain strains generated for this study. However, line 78-80 could be brief out when the parent strain is included in the table.

Response 8: The line 78-80 was revised to “The parent S. cerevisiae yeast strain SEY6210 was used for fluorescence tagging and gene deletion to generate strains for this study” as you suggested in line 106-107.

Comments 9: Line 82-83: Confusing sentence. Include the gene deletion strategy instead.

Response 9: Line 82-83 was revised to “A drug-resistance cassette (hphMX4 or kanMX3) was used to replace the targeted gene in different background strains to generate deletion strains” in line 112-113. The detailed gene deletion strategy was followed up with the following sentences in line 114-121: Either Vph1-GFP tagging or gene deletion was achieved by using polymerase chain reaction (PCR) amplification with corresponding primers in Table 2 to produce DNA fragments, in combination with DNA fragment transformation for endogenous recombination. Vph1-GFP tagged strains were selected on SD-His plates, and gene deletion strains were selected on YPD+Hygromycin B (H8080, Solarbio) or YPD+Geneticin (G8160, Solarbio) depending on the drug-resistance cassette hphMX4 or kanMX3, respectively. The deletion strains were further confirmed with diagnostic PCR with corresponding primers of Table 2.

Comments 10: Line 86: What do you mean by ‘function’ here?

Response 10: “The function of Vtc4 in Vps21 or Snf7 depletion strain” was revised to “The function of Vtc4 depletion in promoting the entry of phagophores into vacuoles in Vps21 or Snf7 depletion strain” in line 122-123.

Comments 11: Line 85 and 94: need rewriting with technical terms rather merely saying described elsewhere.

Response 11: Line 85 was revised to be line 112-113, as stated in Response 9. Line 94 was revised as “All other chemical reagents used in this study were purchased from Sigma (St Louis, MO) or from Amersco (Fair Lawn, NJ), except for SynaptoRed, also known as FM4-64 from Biotium (70021; Hayward, CA), and PCR polymerases and buffers from Takara Biotechnology (Dalian, China)” in line 131-134.

Comments 12: Line 97-102: General microbiology techniques can be removed.

Response 12: Line 97-102 was simplified as “Strains were grown on rich yeast extract peptone dextrose (YPD) plates or in YPD liquid at 26°C for 3 days. If the strains are deletion strains, additional 0.02% Geneticin or 0.03% Hygromycin B were added into medium. YPD liquid contains 2% (w/v) peptone, 1% yeast extract, 2% glucose, with additional 2% agar for plate” in line 136-139.

Comments 13: Line 112: rewrite to improve clarity.

Response 13: Line 110-113 was separated as two sentences and revised as “The final supernatant was removed after centrifugation and replaced with a 60% volume of SD‐N medium to starve. The new culture was incubated for the indicated periods to observe fluorescence images and/or perform immunoblotting assays” in line 148-151.

Comments 14: Section 2.3: Did you normalize your data based on the loading control? If so, explain in methods and Fig 2. Since the loading control is not homogenous as of now, indicate how much protein was added to each lane.

Response 14: We did not normalize our data based on the loading control. We examined and showed loading control to guide the interpretation of data. We usually load 5-10 ul of samples prepared from regular process and adjust the loading sample amount to rerun the gel for 1-2 more times based on the results from previous experiments to have more comparable loading. Yes, the loading control is not homogenous as of now either because of the mutation also affect the expression of G6PDH (we found this is true in some mutants) or the experimental techniques can be improved. For example, the GFP-Atg8 band in vps21∆ cells at 0 h in SD-N is weak, correspondingly, the G6PDH band is weak. We provided more details for crude lysate preparation and loading volume in line 281-282.

For the immunoblotting experiments, we were interested in how much GFP-Atg8 was degraded to GFP. We did not compare the absolute amount of GFP. In contrast, we compared the relative percentage of GFP accounted for the total amount of GFP-Atg8 and GFP. Therefore, no matter the band density was normalized to the loading control or not, the calculated percentage will be the same. We indicated “We did not perform data normalization as we used relative percentage of degraded GFP amount to show the autophagy process and data normalization will not change the percentage” in the figure legend in Figure 2 in line 284-286.

Results:

Comments 15: The authors represented the results in a very neat way with a well detailed figure caption. However, the results section should be strengthened to support observations related to existing literature.

Response 15: We tried to present the results with association to the existing literature, including the description for the proteins or mutants in proper locations.

Comments 16: The authors should explain why a Snf7 deletion background was chosen. What do you mean by “autophagy is normal” in line 159? Also ‘slight’ in line 187 and 189? Remove or explain such vague statements with a proper quantitative or qualitative term/description. The authors should also explain how these single and double mutant strains contribute to the autophagy process or the rationale behind selecting these mutant types.

Response 16: This study was going to explore whether the depletion of VTC complex protein can promote the entry of phagophores into vacuoles in ESCRT mutant cells. Snf7 is a critical subunit of ESCRT-III subcomplex and ESCRT-III is a critical subcomplex for ESCRT. The rationale to select snf7∆ cells and Vtc4 depletion to perform this study was described in line 78-99 in Introduction.

The expression for “autophagy is normal” in line 159 was clarified with details and the whole sentence was revised as “These expecting results indicate that GFP-Atg8 was normally delivered to vacuoles in WT and vtc4∆ cells, but the delivery process was completely inhibited in atg1∆ cells, so that the experimental system is reliable” in line 211-213.  We removed ‘slightly’ in previous line 187 and 189. We added information for how these single and double mutant strains contribute to the autophagy process and the rationale behind selecting these mutant types in proper locations.

Comments 17: In Fig 1, authors also observe the four core strains in early time gaps. Why?

Response 17: To observe more details for core strains between 2 and 8 h of nitrogen starvation as phenotypes in other strains during these durations did not change much.

Comments 18: The scale bars on the microscopy images were almost invisible therefore write down the location in the figure caption to guide the reader.

Response 18: The scale bar size was increased to be visible and the location of scale bar was included in the figure caption.

Comments 19: Line 220: rewrite to improve clarity.

Response 19: Line 220-222 was revised to be “FM4-64 stains both vacuolar membranes and phagophores (Figure 1A and [7]). As phagophores often accumulated next to the vacuolar membranes as clusters, it is hard to distinguish their edges and conclude whether the phagophores entered vacuoles by red fluorescence.” In line 290-293.

Comments 20: Line 228: Does Fig 1 also include data from mCherry tagged strains?

Response 20: We are sorry for the mistake, in previous line 228, “Figures 1 and S2” should be “Figures 3 and S2”as corrected to show in line 299.

Comments 21: Line 229: What do you mean? Where?

Response 21: The line 229-230 was revised to “The indicated strains in Figure 3A were grown and examined as in Figure 1A for red and green fluorescence” as shown in 301-302.

Comments 22: Figure 3: Instead of “another way” use a meaningful title. For example, “mCherry labelled …..” etc.

Response 22: “Another way” was changed to “Using mCherry-Atg8-labeled phagophores” as shown in 301-302.

Discussion:

Comments 23: Many statements require citations.

Line 280-281: Rather than mentioning “as an example”, authors should explain the rationale behind using the double mutant.

Response 23: We tried to include citations for statements. The line 280-281 was changed to “If the depletion of VTC complex protein also promoted the phagophores in ESCRT mutant snf7∆ cells to enter vacuoles under nitrogen starvation, we expected that there would be more phagophores inside vacuoles in snf7vtc4∆ cells than in snf7∆ cells. We compared phagophore localization and degradation in snf7∆ and snf7vtc4∆ cells in this study” to explain the rationale behind using the snf7vtc4∆ double mutant as shown in line 356-360.

Comments 24: Line 287: Explain what is meant by ‘more’ here comparatively.

Response 24: We revised 286-288 as “The mean percentage of GFP degraded from GFP-Atg8 in vps21vtc4∆ and snf7vtc4∆ cells was always higher than that in vps21∆ and snf7∆ cells although without significance (Figure 2)” as shown in line 364-366.

Comments 25: Line 290: Is this due to the florescence degradation or excessive exposure of the film? Could it be the degradation of vtc protein through vacuolar proteases. It has been shown previously that some Vtc proteins are rerouted in the vacuolar lumen by ESCRT and immediately undergo protease degradation. Discuss this with relevant to previous literature on other subunits of VTC complex.

Response 25: The degradation of GFP-Atg8 is highly depending on the cell density, availability of vacuolar hydrolases and the nitrogen starvation time in vps21∆ and ESCRT mutant cells (Figure 1C-F, S1C, S6B, and S6D in previous Ref 11 [Ref15 in revision]). We don’t know how these mutants affect the dynamic ability of hydrolases or the degradation of VTC complex protein through vacuolar proteases. We always controlled the experiment conditions to be the same in the same experiment with strains as the only variation. Under these situations, we can compare the difference in GFP-Atg8 degradation between vps21∆ and vps21vtc4∆ cells or snf7∆ and snf7vtc4∆ cells with the same nitrogen starvation time, as shown in Figure 2. However, when the changes in GFP-Atg8 degradation are minor and the sensitivity of ImageJ measurement always has limitation, the calculated percentage of GFP may be not significant between vps21∆ and vps21vtc4∆ cells or snf7∆ and snf7vtc4∆ cells. All these data are not related to florescence degradation or excessive exposure of the film. We don’t know whether the degradation of VTC complex protein through vacuolar proteases is involved in the degradation of GFP-Atg8. We discussed this with previous literature on other subunits of VTC complex as suggested. Please refer to line 366-386.

Comments 26: Line 300- 303: Authors should have a valid reason rather than a random pick for the selected mutants and explain why other proteins require further investigation.

Response 26: In our previous study in the Ref 8 (Ref 12 in revision), we already showed that most ESCRT mutant cells, especially snf7∆ and vps4∆ cells had the similar and strongest phenotypes for phagophore accumulation.   In our previous study in Ref 11 (Ref 15 in revision), we already showed that the depletion of Vtc1, 2, 4, and 5 promoted the entry of phagophores into vacuoles in vps21∆ cells similarly. Then we chose snf7∆ cells to represent ESCRT mutant cells, and vtc4∆ to represent the depletion of any one of Vtc1, 2, 4, and 5. It is rationale to expect that the depletion of Vtc1, 2, and 5 in snf7∆ or the depletion of Vtc1, 2, 4, and 5 in vps4∆ will generate similar results as in snf7vtc4∆ cells, although the real result need experiments to support. Please refer to line 389-403.

Comments 27: Line 306: What is 2018-NC?

Response 27: We are sorry for the mistake. It is an indicate to insert the reference published in Nature Communications in 2018 with the title “An autophagy assay reveals the ESCRT-III component CHMP2A as a regulator of phagophore closure.” Corrected in line 405. Thanks.

Comments 28: The authors should be able to discuss their findings in relation to past findings to provide the significance of this work.

Response 28: This study mainly broadens the findings in our previous study in Ref 11 (Ref 15 in revision) and suggest a general role of VTC complex proteins in negatively regulating other cargoes outside or on vacuolar membranes to enter vacuoles in yeast. Please refer to line 416-418 and line 427-428.

Comments 29: If Vtc4 depletion promotes phagophores entry, do you think that vps21 and vtc4 act in a dependent way? Explain.

Response 29: We found Vtc4 depletion obviously promoted phagophores entry into vacuoles in vps21∆ in Ref 11 (Ref 15 in revision) and in snf7∆ cells in this study. We also noticed that the GFP-Atg8 degradation in vtc4∆ cells was also more than that in wild-type cells in mean value in Figure 2. We don’t think that Vps21 and Vtc4 act in a dependent way, but the obvious phenotype of phagophore accumulation in vps21∆ cells provided a convenient way to study whether VTC complex protein depletion promoted phagophore entry. We discussed them in the last paragraph. Please refer to line 418-426.

Round 2

Reviewer 1 Report

I accept authots answers to my questions as a reviewer.